# Metabolic Pathway Modeling in Muscle of Male Marathon Mice (DUhTP) and Controls (DUC)—A Possible Role of Lactate Dehydrogenase in Metabolic Flexibility

**DOI:** 10.3390/cells12151925

**Published:** 2023-07-25

**Authors:** Julia Brenmoehl, Elli Brosig, Nares Trakooljul, Christina Walz, Daniela Ohde, Antonia Noce, Michael Walz, Martina Langhammer, Stefan Petkov, Monika Röntgen, Steffen Maak, Christina E. Galuska, Beate Fuchs, Björn Kuhla, Siriluck Ponsuksili, Klaus Wimmers, Andreas Hoeflich

**Affiliations:** 1Institute of Genome Biology, Research Institute for Farm Animal Biology (FBN), Wilhelm-Stahl-Allee 2, 18196 Dummerstorf, Germany; 2Department of Neurology, Neuroimmunological Section, University Medicine Rostock, Gehlsheimer Str. 20, 18147 Rostock, Germany; 3Department of Animal Genomics, Centre for Research in Agricultural Genomics (CRAG), Campus de la Universitat Autònoma de Barcelona, 08193 Cerdanyola, Spain; 4Lab Animal Facility, Institute of Genetics and Biometry, Research Institute for Farm Animal Biology (FBN), Wilhelm-Stahl-Allee 2, 18196 Dummerstorf, Germany; 5Institute of Muscle Biology and Growth, Research Institute for Farm Animal Biology (FBN), Wilhelm-Stahl-Allee 2, 18196 Dummerstorf, Germany; 6Core Facility Metabolomics, Research Institute for Farm Animal Biology (FBN), Wilhelm-Stahl-Allee 2, 18196 Dummerstorf, Germany; 7Institute of Nutrition, Research Institute for Farm Animal Biology (FBN), Wilhelm-Stahl-Allee 2, 18196 Dummerstorf, Germany

**Keywords:** glycogen, lactate, pathway analysis, skeletal muscle, treadmill training

## Abstract

In contracting muscles, carbohydrates and fatty acids serve as energy substrates; the predominant utilization depends on the workload. Here, we investigated the contribution of non-mitochondrial and mitochondrial metabolic pathways in response to repeated training in a polygenic, paternally selected marathon mouse model (DUhTP), characterized by exceptional running performance and an unselected control (DUC), with both lines descended from the same genetic background. Both lines underwent three weeks of high-speed treadmill training or were sedentary. Both lines’ muscles and plasma were analyzed. Muscle RNA was sequenced, and KEGG pathway analysis was performed. Analyses of muscle revealed no significant selection-related differences in muscle structure. However, in response to physical exercise, glucose and fatty acid oxidation were stimulated, lactate dehydrogenase activity was reduced, and lactate formation was inhibited in the marathon mice compared with trained control mice. The lack of lactate formation in response to exercise appears to be associated with increased lipid mobilization from peripheral adipose tissue in DUhTP mice, suggesting a specific benefit of lactate avoidance. Thus, results from the analysis of muscle metabolism in born marathon mice, shaped by 35 years (140 generations) of phenotype selection for superior running performance, suggest increased metabolic flexibility in male marathon mice toward lipid catabolism regulated by lactate dehydrogenase.

## 1. Introduction

Energy supply during physical exercise relies on the involvement of phosphagen, glycolytic, and oxidative energy pathways. Whether energy (ATP) is generated more through oxygen- and mitochondria-independent anaerobic or oxidative energy fluxes depends on the intensity and duration of physical activity, training status, nutrition, sex, age, and ambient conditions [1]. Strength training with short bursts of maximal work, such as supramaximal sprinting or field sports, relies primarily on phosphocreatine and glycogen catabolism associated with anaerobic glycolysis since these processes have high ATP production rates [2]. In contrast, continuous endurance activities, such as running, cycling, or swimming, rely primarily on oxidative (aerobic) phosphorylation of carbohydrates and fatty acids, providing a higher ATP production capacity [2]. During the early stages of aerobic exercise, muscle glycogen and intramuscular fatty acids are oxidized, and, with prolonged duration, peripheral glucose and fatty acids are taken up and metabolized [3].

Thus, substrates for anaerobic and aerobic glycolytic metabolism can be glycogen, the glucose reservoir stored in muscle fibers, or glucose circulating in the blood, taken up by muscle via constitutive glucose transporters 1 (GLUT1) or GLUT4 transporters, which are mobilized from intracellular storage vesicles to the plasma membrane after insulin stimulation. The anaerobic glycolysis product lactate, converted from pyruvate by lactate dehydrogenase, oxidizing NADH, is produced primarily in glycolytic fibers [4] and is increasingly exported from cells with the onset of muscle contraction [5]. According to the cell–cell lactate shuttle concept [6], lactate serves as a primary energy substrate for oxidative muscle fibers within a working muscle [7] or other lactate-consuming organs such as the heart, brain, liver, kidneys, and adipose tissue [8]. Lactate uptake occurs when lactate has been metabolized in the cell, and the lactate breakdown rate exceeds the rate of lactate production [9]. Thus, the blood lactate level reflects the balance between lactate production and degradation.

Lactate export and import are accomplished by specific bidirectional monocarboxylate transporters (MCTs) in symport with protons [10], with MCT4 responsible for lactate export from glycolytic fibers [11] and MCT1 for the import into type I fibers, enabling the lactate uptake for glycolytic oxidation [12]. In this process, lactate is first converted to pyruvate by cytosolic lactate dehydrogenase, which is then transported into the mitochondria by mitochondrial pyruvate carriers 1 (MPC1) and 2 (MPC2) [13]. It has also been described that lactate can be transported directly into mitochondria via mitochondrial MCT1 transporters and then converted by mitochondrial lactate dehydrogenase to pyruvate [14], which is then further metabolized and fed into the TCA cycle and respiratory chain to generate ATP for muscle contractions. However, this concept of intracellular lactate shuttling [6] is controversially discussed due to methodological discrepancies in proteolytic tissue processing, according to Cruz et al. [15].

Quantitatively more ATP molecules are produced during fatty acid oxidation (FAO). Fatty acids originate from intramuscular and peripheral fat depots. Transport of fatty acids into the muscle cell occurs by diffusion or is facilitated by inducible CD36 together with fatty-acid-binding proteins (FABPs) or the fatty acid transport protein FATP [16]. Fatty acids are activated in the muscle cell cytosol before being transported into mitochondria for complete oxidation.

Exercise intensity at the muscle level defines the substrate source for energy production [1,17]. During continuous cycling with a workload of 30% maximal oxygen uptake, glucose and free fatty acids contribute to substrate utilization. Glucose oxidation predominates for up to 90 min and is then surpassed by FAO [18]. Fatty acid oxidation is increased during low- and moderate-intensity endurance exercise, whereas carbohydrate oxidation is also preferred during marathons [19]. Maximum fat oxidation occurs at 64% maximal oxygen uptake [20], whereas during exercise above 80% maximal oxygen uptake, fat oxidation decreases, and plasma glucose and muscle glycogen utilization predominate [17]. The physiological adaptability to respond appropriately to increasing energy demands and fuel availability (e.g., in response to exercise or fasting) at the level of substrate recognition, transport, utilization, and storage level is indicative of high metabolic flexibility [21,22]. Training-associated metabolic flexibility is regulated by transcription factors involved in the transcription of metabolic genes relevant to oxidative phosphorylation, glucose and lipid transport, or mitochondrial biogenesis [22]. A targeted and timely switch to the oxidative degradation of energy-rich fatty acids could not only contribute to the long-term energy supply of the muscle, but also positively affect the mobilization of storage lipids.

The marathon mouse model DUhTP, generated by long-term paternal selection for high treadmill performance [23], is characterized by superior forced running performance but also by increased lipid accumulation. In response to voluntary moderate physical activity, increased lipid turnover has been observed in the liver [24,25] and adipose tissue [26], particularly subcutaneous fat [26,27], suggesting greater metabolic flexibility toward lipid metabolism in this mouse model. In this work, we used the marathon mice and a control strain, both derived from the same genetic background, to perform three weeks of treadmill training at an intensive speed that favors glycolytic utilization. We hypothesized that the marathon mice rely on a lipid-based metabolism despite the high speed and investigated the differences in energy metabolism in the muscles of both strains at the molecular and biochemical levels. Muscle energy metabolism was compared between marathon mice and unselected controls (genetic model), between physically active and sedentary animals of one strain (experimental model), and between trained animals of both strains (genetic × experimental model), respectively, to identify selection-related adaptations in carbohydrate- and especially lipid-based energy metabolism and to further relate them to the increased lipid turnover in subcutaneous fat observed in marathon mice. The present study discusses the results of molecular pathway modeling in muscle concerning systemic or global biomarkers of energy metabolism.

## 2. Materials and Methods

### 2.1. Animals and Experimental Design and Sample Collection

All animal experiments were performed in the mouse facility of the Research Institute for Farm Animal Biology in Dummerstorf, Germany. The experiments were approved by our internal institutional review board and were conducted following national and international animal welfare guidelines (Animal Welfare Act (TierSchG); AZ 7221.3.1.014/17 and AZ 7221.3-1-064/19). We used the non-inbred mouse line DUC, generated in the 1970s from a polygenic base population without phenotype selection [28], and the selection line DUhTP [23]. This line was paternally selected for high treadmill performance starting eleven years later from the unselected control line for about 140 generations and is now maintained as a conservation line. A recently published study describes the selection procedure in more detail [29]. Nevertheless, in brief, in each generation, males completed a submaximal run after mating, avoiding inbreeding, to identify the best runners and select their offspring as the next generation’s parents. This generated a non-inbred mouse line characterized by exceptional running ability without prior training. After around 140 generations, the DUhTP line showed a 3–4 times longer running distance than the unselected control.

Male animals of generations 141 and 151 (DUhTP) and 185 and 196 (DUC) were used in this study. The animals were housed in the SPF barrier unit in polysulfone cages of 267 × 207 × 140 mm (H-Temp PSU, Type II, Eurostandard Tecniplast, Hohenpeißenberg, Germany) in compliance with hygiene management and health surveillance according to FELASA recommendations. The housing conditions of the mice were defined by a 12 hlight-dark cycle (room temperature = 22.5 °C ± 0.2 °C, humidity = 45–60%), and the animals had free access to autoclaved Ssniff^®^ M-Z food (12 kJ% fat, 27 kJ% protein, 61 kJ% carbohydrates; gross energy 16.7 MJ/kg; Ssniff-Spezialdiäten GmbH, Soest, Germany) and water.

After weaning, males of both lines were kept in pairs in one cage until 48 days of age, when they were randomly divided into two groups and housed in individual cages. From 49 to 70 days of age, a part (n = 16 per generation) of the animals completed a fixed 3-week exercise program (“trained”) [30] on a computer-controlled treadmill (TSE company), while the remaining animals (n = 10) remained in their cages and served as a control group (“sedentary”, “sed”). Food consumption and body weights of the mice during this experiment are summarized in the Appendix A. Since the mice are characterized by a different running performance [31], we adapted the training duration to the running capacity of each strain in order to challenge the animals equally. We based this on the completed submaximal temporal running performance of previous generations (DUhTP: 133 min, DUC: 66 min) [31]. We chose approximately 23% of the previous average submaximal running duration as the training duration and accordingly estimated 30 min and 15 min per training session for the DUhTP and control males, respectively. The training was conducted during the light phase between 8 and 9 am, five days per week for three weeks. After an initial run and a two-day rest, regular training (5× per week, 15/30 min, 0% incline) was started. For this, the mice were exposed to an initial speed of 0.2 m/s for 20 s, then to a speed of 0.36 m/s for 160 s, followed by a final speed period. The final speed up to 0.5 m/s (DUhTP) and 0.42 m/s (DUC) was reached after a series of gradual weekly increases over time (detailed study design published in [30]). If an animal did not perform on the motorized treadmill, the run was stopped for that animal. If the same animal also did not perform the next day, it was excluded from the experiment. Animals with possible injuries were also excluded from the experiment.

At 70 days of age, the animals were sacrificed by decapitation either after the last running session or in a sedentary state. Thus, we ensured that the influence of the current treadmill run with the background of three weeks of training could be investigated. Plasma or serum was collected from the trunk blood and stored at −20 °C. Tissues (liver, rectus femoris muscle, pituitary gland, epididymal, and posterior subcutaneous fat) were removed, weighed, shock-frozen in liquid nitrogen, and stored at −70 °C for subsequent analysis.

### 2.2. Immunohistochemical Evaluation of Musculus rectus femoris

Immunohistochemistry was performed on the *Musculus rectus femoris* (Mrf) of seven (DUC sed) to eight animals per group (DUhTP sed, DUhTP trained, DUC trained), as described by Rehfeldt et al. [32]. Briefly, 10 µm cross-sections were cut at the end of the proximal origin tendon, and the metabolic fiber types (red, intermediate, white) were stained using NADH-tetrazolium reductase. The muscular cross-sectional area (CSA), and the fibers over an area of 3.316 mm^2^ (approximately 32% of the total CSA) were determined separately by image analysis (TEMA v1.00, Scan Beam APS, Hadsund, Denmark). In these analyzed areas, a total of four fields of view (representative images for one animal per group are shown in Appendix A), 100–150 muscle fibers each in the light (superficial) and dark (profound) areas were evaluated for the distribution and CSA of the red, intermediate, and white fibers. Fat content was analyzed by using Oil Red O (Sigma-Aldrich, Taufkirchen, Germany). Therefore, the 10 µm cross-sections were fixed in 4% formal calcium fixative for 5 min, washed in aqua dest. for another 5 min, incubated with Oil Red O (0.5% wt/vol dissolved in 99% isopropanol and diluted with 4/6 parts H_2_O before use), and washed in aqua dest., both for another 15 min.

### 2.3. RNA Isolation, Next-Generation Sequencing (NGS), and Differential Gene Expression Analysis

Total RNA was extracted from Mrf tissue samples using Trizol (Sigma-Aldrich) and treated with DNase I to eliminate DNA contamination. Testing of RNA quality, enrichment and fragmentation of RNA, reverse transcription into cDNA, and DNA library generation were performed as previously described [30]. The normalized multiplexed DNA libraries with a 0.5% PhiX control were amplified clonally clustered with the cBot system and then sequenced at 2 × 101 bp in high-performance mode on a HiSeq2500 (Illumina, San Diego, CA, USA) in our sequencing facility at the Research Institute of Farm Animal Biology (FBN). Raw sequencing reads (fastq) were processed and mapped using Hisat2 and the mouse reference genome (GRCm38) as previously published [30]. The resulting gene count data (ArrayExpress accession E-MTAB-12072) were further analyzed for differentially expressed genes (DEGs; Appendix A) using edgeR and R dependency packages. Genes with low readings (count per million (cpm)) were excluded to obtain only genes with >0.5 cpm in at least four libraries. The edgeR standard parameters were applied with the option trimmed mean of M values (TMM) considering library size and composition bias and the option estrimateGLMRobustDisp to estimate interlibrary variation. The glmFit and glmLRT functions implemented in edgeR were used for statistical testing of DEGs.

### 2.4. Validation by Fluidigm

RNASeq data were validated by 2-step reverse transcription-quantitative PCR using the Fluidigm technique [33] described in Walz et al., 2021 [30] with the following modifications. Nine DEGs with different read counts (cpm) were selected for this purpose; the primers used are listed in Appendix A. For reverse transcription, 800 ng of RNA were used, and the cDNA concentration in the PCR to check primer quality ranged from 25 to 0.0125 ng. We calculated the relative expression of each primer-template combination and related it to four housekeeping genes (Actb, Pgk1, Rplp2, Rpl26) using Data Analysis Gene (DAG) expression software [34]. Group-independent correlation analysis was performed for each gene between RNA sequencing signals (cpm) and relative qPCR expression (2^−ΔΔCt^) (Appendix A) after checking for outliers (ROUT method, Q = 0.1%) in GraphPad Prism V 9.4.1. Pearson’s correlation coefficient and *p*-value were calculated and displayed.

### 2.5. Pathway Analysis

Significantly upregulated and downregulated DEGs (FDR < 0.05, Appendix A) of the comparison groups DUhTP sed versus (vs.) DUC sed, DUhTP trained vs. DUC trained, DUhTP trained vs. DUhTP sed, and DUC trained vs. DUC sed were bioinformatically analyzed as previously described [30]. In order to understand which pathways were activated or silenced, enrichment analysis was performed separately for up- and downregulated genes of the DUhTP trained vs. DUC trained comparison (FDR < 0.05). For this purpose, the DAVID V6.8 software [35] was used with default threshold filters (EASE = 0.1, minimum gene count = 2; access date: 25 November 2022). To ensure stringency, the results of the enrichment analysis (KEGG pathways) were subjected to a multiple testing correction (adjusted *p*-value Benjamini < 0.05). The obtained list of KEGG pathways for the upregulated and downregulated DEGs is summarized in Appendix A, including the minimum gene number for each KEGG pathway. Since 423 significantly upregulated DEGs (Appendix A) are involved in the regulation of metabolic pathways, the significant DEGs (FDR < 0.05) from all comparison groups were entered into the Pathview software [36] to obtain graphical visualization of their regulation in the glycolysis, beta-oxidation, TCA cycle, and oxidative phosphorylation pathways (access date: 1 December 2022).

### 2.6. Determination of Enzyme Activities

Frozen muscle tissue samples (100 mg/per mouse) were powdered under liquid N_2_ using a mortar and pestle. All powders were homogenized in a 1:20 (weight/volume) dilution of 0.01 M potassium phosphate buffer containing 150 mM potassium chloride, 5.8 mM monopotassium phosphate, 4.2 mM dipotassium phosphate, and 1 mM EDTA (pH = 6.9) and centrifuged (14,000× *g*, 15 min, 4 °C). The supernatants were used for further analysis. Creatine kinase (EC 2.7.3.2.) activity was measured at 37 °C in 1:400 diluted muscle homogenates using a commercial kit (CK-NAC-Hit kit, IFCC method, BIOMED Labordiagnostik GmbH, Oberschleißheim, Germany) according to the manufacturer’s instructions. Enzyme activity of lactate dehydrogenase (LDH; EC 1.1.1.28) and isocitrate dehydrogenase (IDH; EC 1.1.1.42) was determined by using 0.19 mM NADH and 0.757 mM sodium pyruvate or 4 mM isocitrate, 3.3 mM MnSO_4_, and 0.35 mM NADP, respectively, according to an adapted protocol (http://www.sigmaaldrich.com/life-science/metabolomics/enzyme-explorer/learning-center/assay-library.html, accessed on 25 September 2019). The protein suspension was used 1:20 diluted (LDH) or undiluted (IDH), respectively. All enzyme activities were determined on a Spectramax Plus384 spectrophotometer/plate reader (Molecular Devices Corporation, Sunnyvale, CA, USA) in technical triplicates per animal.

Furthermore, pyruvate dehydrogenase activity was evaluated using a kit from Abcam (ab109902, Cambridge, UK). For this purpose, protein lysates were isolated from 50 to 100 mg Mrf tissue powder according to the manufacturer’s instructions, which were uniformly adjusted to 8 µg/µL muscle protein concentration in deviation from the instructions. Proteins were extracted by adding the detergent solution at a final dilution of 1/20. For each sample, 250 µg protein per well was added to the plate and incubated for three hours at room temperature. After recording the linear kinetics by the increase in absorbance in 30 min, the rate between the two time points (according to the instructions) was calculated and used to determine the PDH activity relative to the DUC sed rate (100%).

### 2.7. Determination of Plasma Glucose, Insulin, and Lactate

Glucose and insulin were determined in plasma using commercial kits (Glucose GOD-PAP:LT-GL 0251, Labor&Technik Eberhard Lehmann, Berlin, Germany; Ultra Sensitive Mouse Insulin ELISA Kit, Crystal Chem, Zaandam, The Netherlands) according to the manufacturer’s protocol, with a reduced total approach for glucose. Plasma lactate levels were analyzed spectrophotometrically using a semi-automated analyzer (Abx Pentra 400, Horiba, Kyoto, Japan) and a commercial kit for L-lactate concentrations (A11A01721, Horiba).

### 2.8. Untargeted LC-MS/MS Analysis of Lipids and Fatty Acids in Serum

A total of 30 µL of serum was mixed with 120 µL of isopropanol for lipid extraction and 120 µL of methanol for small metabolite extraction, vortexed for 15 s, and centrifuged for 15 min at 4 °C and 15,800× *g*. The supernatant was transferred to an appropriate sample vial for further analysis. A QC pool was created from all extracted serum samples. After equilibration of the system with the QC samples, the samples were randomly analyzed using an ultra-high-performance liquid chromatography-tandem mass spectrometry (UHPLC-MS/MS) system (Vanquish UHPLC-system with heated electrospray ionization (HESI) QExactive plus Orbitrap mass spectrometer (Thermo Scientific, Waltham, MA, USA)) in the positive and negative ionization modes. MS data were acquired over a scan range of 100–1200 *m*/*z* with a full MS resolution of 70,000 and a data-dependent MS2 resolution of 17,500. Chromatographic separation of the lipids was performed on a reversed-phase column (Accucore Polar Premium 100 × 2.1 mm (2.6 µm) with guard column: Accucore Polar Premium 10 × 2.1 mm (2.6 µm); Thermo Scientific). The autosampler temperature was set to 10 °C and maintained throughout the measurements. Mobile phase A consisted of 60% acetonitrile, 10 mM ammonium formate, and 0.1% formic acid in ultrapure water, and mobile phase B consisted of 90% isopropanol, 10 mM ammonium formate, and 0.1% formic acid in ultrapure water. The gradient was 20–100% B in 8.5 min over a total run time of 15 min. The flow rate employed was 0.4 mL/min, and the column temperature was 55 °C. Chromatographic separation of the small metabolites was performed on a reversed-phase column (Accucore Polar Premium 100 × 2.1 mm (2.6 µm) with guard column: Hypercarb 10 × 2.1 mm; Thermo Scientific) at 45 °C. The temperature of the autosampler was 6 °C. Mobile phase A was 0.1% formic acid in ultrapure water, and mobile phase B was 0.1% formic acid in methanol. The components were separated at a 0.5 mL/min flow rate using the following gradient: 1–99% B in 12 min. Annotation and relative quantification of individual metabolic and lipid species were performed by Compound Discoverer 3.1 (Thermo Scientific) and Lipid Data Analyzer software from the University of Graz (Austria).

### 2.9. Determination of Muscular Glycogen Content

The quantitative muscle glycogen determination is based on Hassid and Abraham [37] and measures glucose, as reduced sugar, colorimetrically at 680 nm. For this purpose, 25–50 mg of muscle tissue were boiled for 20 min in 800 mL 30% KOH, cooled on ice, mixed with 1 mL 95% ethanol, brought to boiling, cooled again, and centrifuged for 15 min (750× *g*). The resulting pellet was dissolved in 1 mL H_2_O, mixed with 1 mL ethanol, and centrifuged again. The pellet was then dissolved in 500 mL H_2_O and 1 mL 0.2% anthrone reagent (0.2 g anthrone/100 mL 95% sulfuric acid) and boiled (10 min). After cooling, the solution was measured at 680 nm. Four determinations were performed per animal (n = 8) of each group, and the values were recalculated using a glucose standard series between 2–100 µg/mL or 10–500 mg/g sample weight. Glycogen content is expressed as mg/g wet tissue.

### 2.10. Statistical Analysis

Data analysis and graphs were performed using GraphPad Prism 9.4.1 (GraphPad Software, San Diego, CA, USA). The data could be considered as approximately normally distributed; outlier identification was performed using the GraphPad Prism 9.4.1 statistical analysis package. Different animal numbers were used for each analysis. Therefore, the sample number is noted in the figure legend for the respective result. All data were analyzed using two-way ANOVA with multiple comparisons, considering the two different mouse strains (DUhTP, DUC) and the two different treatments (sed, trained). The distribution of fiber types or areas of fiber types in the superficial and deep regions of mice’s Mrfs were statistically evaluated using the linear model (lm in R) followed by the TukeyHSD post hoc test (TukeyHSD in R) to test the dependence between phenotype and the interaction of strain × treatment × muscle fiber type. The effect of training was calculated using the unpaired *t*-test with Welch correction. Data were presented as scatter plots with mean and standard deviation. Pearson correlation coefficients were calculated using GraphPad Prism 9.4.1 to determine the relationships between muscle mass and cross-sectional area (CSA). The effects and differences were considered significant at *p* < 0.05.

## 3. Results

### 3.1. Reduced Body and Muscle Weight in Response to Selection and Training in Marathon Mice

At 70 days of age, DUhTP mice had a significantly lower body weight than control mice, independent of training (*p* < 0.0001; Figure 1a). Physical activity decreased body weight (*p* < 0.01), as well as fat mass (*p* < 0.005), in particular epididymal (Figure 1b; *p* < 0.05) and posterior subcutaneous fat (*p* < 0.06), exclusively in DUhTP mice. Femoral muscle weights were significantly lower in sedentary DUhTP than in DUC mice (*p* < 0.01; Figure 1c). After three weeks of training, femoral muscle weight was reduced in DUhTP but not in DUC mice (Figure 1c, −10.8%, *p* < 0.05). As a result, femoral muscle weight differed by approximately 28% (*p* < 0.0001) between trained animals of both lines. However, the relative femoral muscle weight to body mass was approximately 1.28% in all groups.

In sedentary conditions, the cross-sectional area (CSA) of the Mrf was similar between the two lines, whereas in trained DUhTP mice, the CSA of the Mrf was smaller than in trained DUC mice (Figure 1d, −17%, *p* < 0.05). Interestingly, muscle mass and CSA were positively correlated in DUC (sed: Pearson coefficient r = 0.9; trained: r = 0.83, *p* < 0.005) but not in DUhTP mice. The number of fibers per mm^2^ analyzed muscle region (see Method part) was not significantly different in sedentary mice of both lines (Figure 1e, DUhTP: 428.7 fibers/mm^2^; DUC: 360.6 fibers/mm^2^; *p* = 0.087). In contrast, trained DUhTP mice had significantly higher fiber numbers per mm^2^ than trained DUC animals (434.7 fibers/mm^2^ vs. 349.8 fibers/mm^2^; *p* < 0.05; Figure 1e). Interestingly, regardless of the training status, the Mrfs of DUhTP mice appeared macroscopically darker (Appendix A) and had significantly more mitochondria-rich fibers, i.e., more red and intermediate fibers, per mm^2^ analyzed muscle region than DUC mice (*p* < 0.01; Figure 1f). In trained DUhTP’s Mrfs, significantly more mitochondria-rich fibers were found than in trained DUC mice (*p* < 0.05). Intramuscular fat content in Mrf was not significantly different between sedentary and trained DUhTP and DUC mice (Appendix A). However, an exercise-induced reduction in intramuscular fat content was observed in both lines (*p* < 0.05). The distribution of oxidative (red), glycolytic (white), and hybrid (intermediate) fibers in both superficial and profound areas of the Mrf (Appendix A) was examined in all experimental groups. In both lines, significantly more glycolytic than intermediate or red fibers were detected in the superficial muscle region (*p* < 0.0001; Appendix A). In the deep muscle region, more red (approximately 41%) than intermediate (approximately 29%) or white fibers (approximately 30%) were found in sedentary and trained DUhTP and sedentary DUC mice, respectively (*p* < 0.05; Appendix A). In trained DUC mice, the fiber type percentage was similar (red: 36.3%, intermediate: 30.3%, white: 33.4%). In trained and sedentary animals of both lines, the average glycolytic fiber area was significantly larger than the intermediate or red fiber area in superficial and profound muscle regions (*p* ≤ 0.05; Appendix A). Control animals had a higher mean glycolytic fiber area than DUhTP animals in both training conditions in the deep muscle region (*p* < 0.01; Appendix A).

Thus, the results show that long-term selection for high treadmill performance results in a phenotype with lower body and muscle mass weight, which under training conditions is further characterized by lower Mrf muscle cross-sectional areas, higher muscle fiber number per mm^2^, and more mitochondria-rich muscle fibers. Exclusively in the DUhTP mice, a decrease in fat depots was also observed by training, consistent with the hypothesis of a predominant role of lipids in muscle metabolism.

### 3.2. Analysis of mRNA Indicates Induction of Glycolysis, Beta-Oxidation, TCA-Cycle, and Oxidative Phosphorylation in Muscles from DUhTP Mice

To investigate muscle gene expression and metabolic pathways in response to selection and physical activity, we performed NGS using total RNA preparations isolated from trained and sedentary DUhTP and DUC Mrfs. Differentially expressed genes (DEGs) in different experimental groups were characterized by a false discovery rate (FDR) of <0.05. Accordingly, 13,982 DEGs were identified and quantified in muscle from the different experimental groups (data set NGS = Appendix A). Direct mouse strain comparison (DUhTP sed vs. DUC sed) identified 5916 transcripts differentially regulated due to long-term selection for high-endurance exercise (Appendix A). These transcripts were upregulated and downregulated to a similar extent. After three weeks of training, the total number of altered muscle transcripts between the two lines increased to 7042, with 460 more downregulated than upregulated transcripts. Based on the number of DEGs, exercise had a stronger effect on gene expression in DUhTP (DUhTP trained vs. DUhTP sed: 1606 DEGs) than in DUC mice (DUC trained vs. DUC sed: 130 DEGs). The comparison between trained DUhTP and trained DUC muscle transcripts revealed that several signaling pathways were silenced or less activated (Appendix A, downregulated), whereby multiple metabolic pathways were highly activated (Appendix A, upregulated).

Therefore, all DEGs identified in DUhTP sed vs. DUC sed, DUhTP trained vs. DUC trained, DUhTP trained vs. sed, and DUC trained vs. sed were implemented into Pathview to obtain their visualization (Appendix A) in some metabolic KEGG pathways enriched by DAVID. Pathway analysis suggested higher glycolysis activity in sedentary DUhTP mice than in sedentary DUC mice. Indeed, several enzyme transcripts catalyzing the conversion of glucose-6-phosphate to pyruvate and further to acetyl-CoA were increased (Appendix A, Table 1). However, the gene expression of hexokinase (2.7.1.1), the initial enzyme of glycolysis, was lower in DUhTP than in DUC mice. The expression of glucokinase (2.7.1.2), which plays a role in the blood glucose sensing system, was also lower in marathon mice. Training increased hexokinase transcription in both DUhTP and DUC animals such that significant differences in hexokinase transcription between the two exercised strains could no longer be detected. However, the reduction in glucokinase expression persisted in trained DUhTP mice compared with trained control animals. In addition, fewer transcripts of ADP-dependent glucokinase (2.7.1.147) were also found in the muscles of trained DUhTP animals (FDR < 0.0007). In DUhTP mice, physical activity increased the transcript levels of glyceraldehyde-3-phosphate dehydrogenase (1.2.1.12). Interestingly, transcripts encoding glycogen-degrading enzymes were also significantly decreased (FDR < 0.05) in DUhTP compared with control muscles (Table 1). Transcripts for phosphorylase b kinase, glycogen debranching enzyme, and phosphoglucomutase-2 were decreased in muscles of sedentary and trained DUhTP compared with the corresponding DUC mice (FDR < 0.05). In addition, lower transcript levels of UTP-glucose-1-phosphate uridylyltransferase (FDR < 0.015) but higher glycogen phosphorylase (FDR < 0.002) and glycogen synthase transcripts (FDR < 0.03) were detected in DUhTP muscles compared with control muscles after training. For fatty acid degradation, more enzyme transcripts (FDR < 0.05) were found in the Mrf of DUhTP compared with control animals (Appendix A, Table 1). High abundances of Cpt2 (FDR < 0.004), acyl-CoA dehydrogenases (1.3.8.7, 1.3.8.9, FDR < 0.002), enoyl-CoA hydratase (4.2.1.17, FDR < 0.04), 3-hydroxyacyl-CoA dehydrogenase (1.1.1.35, FDR < 0.0001), and 3-ketoacyl-CoA thiolase (2.3.1.16, FDR < 0.0002) were detectable in marathon mice (Table 1). In response to treadmill running, the expression of these genes was further increased. Furthermore, increased expressions of another acyl-CoA dehydrogenase (1.3.8.8, FDR < 0.0001) and an acyl-CoA oxidase (1.3.3.6, FDR < 0.05) were detected in trained DUhTP muscles compared with those of controls. Transcripts of the long-chain fatty acid CoA ligase (6.2.1.3) were lower in DUhTP mice’s muscles than in controls (FDR < 0.0001), independent of exercise. The number of transcripts corresponding to citrate cycle enzymes was significantly higher in Mrf of DUhTP than in control mice (FDR < 0.05), independent of training status (Appendix A, Table 1). However, physical activity enhanced these significant differences, although no significant expression differences were detected within either line. Substantial expression differences were found for the enzymes isocitrate dehydrogenase (1.1.1.41; 1.1.1.42, FDR < 0.002), succinate-CoA ligase (6.2.1.4; 6.2.1.5, FDR < 0.009), succinate dehydrogenase (1.3.5.1, FDR < 0.0007), and malate dehydrogenase (1.1.1.37, FDR < 0.0001) (Table 1). The increase in the expression of alpha-ketoglutarate dehydrogenase was significant after the last training session (FDR < 0.0001). In addition, many oxidative phosphorylation pathway genes were upregulated in DUhTP mice compared with controls, independent of physical activity (Appendix A). When compared with sedentary animals, many transcripts of subunits of complexes I, II, III, IV, and V were upregulated in DUhTP muscles. Interestingly, after three weeks of training, the mitochondria-encoded subunits of complex I (ND1, ND2, ND3, ND4L) were significantly reduced between trained DUhTP and DUC mice (FDR < 0.008, Appendix A). While no changes were detected in DUC mice after physical activity (DUC trained vs. DUC sed), DUhTP animals showed significant downregulation of mitochondria-encoded complex I subunit transcripts (DUhTP trained vs. DUhTP sed, FDR < 0.03).

Thus, the muscle transcriptome of the marathon mice, in contrast to that of the controls, shows upregulation of gene expression of glycolysis, FAO, TCA cycle, and respiratory chain enzymes but downregulation of glycogen degrading enzymes, indicating higher metabolic activity in the muscle of the selection line.

### 3.3. Plasma Levels of Energy Metabolites in Response to Phenotype Selection and Physical Activity

To challenge the bioinformatics pathway analysis predictions, we assessed selected plasma levels of energy metabolites in all experimental groups. Sedentary DUhTP and DUC mice had similar plasma levels of glucose (Figure 2a), lactate (Figure 2b), and insulin (Figure 2c). In response to exercise after three weeks of training, plasma glucose and plasma lactate levels were increased in DUC mice (*p* < 0.05). Insulin levels were not significantly altered by exercise in either strain. However, an exercise-induced decrease in plasma insulin levels was observed (*p* < 0.01; Figure 2c). Whereas exercise had no effect on plasma glucose or lactate concentrations in trained DUhTP mice, trained DUC mice were characterized by higher glucose and lactate levels than trained DUhTP mice (*p* < 0.05).

Interestingly to note, three days after treadmill training, serum lactate levels were significantly reduced in the DUhTP animals (2.78 ± 0.42 mmol/L) compared with sedentary DUhTP mice (4.96 ± 0.45 mmol/L, *p* < 0.04). In contrast, in DUC mice, lactate levels three days after exercise were similar (4.76 ± 0.90 mmol/L) to those in sedentary DUC mice (4.77 ± 1.58 mmol/L).

Since lactate is an effector of lipolysis, we performed a detailed analysis of lipids in mouse blood samples. Serum from trained and sedentary DUhTP mice was characterized by significantly increased concentrations of specific circulating lipid species (acylcarnitines, di/triglycerides, and fatty acids derivates) compared with corresponding control mice (Figure 3, top left and right panel, Appendix A). In particular, increased levels of acylcarnitines, phospholipids, and fatty acid derivatives were found in trained DUhTP mice compared with sedentary DUhTP mice (Figure 3, lower left panel, Appendix A). In addition, during exercise, the levels of certain phospholipids that act as signaling molecules increased in DUhTP and control mice, but with more pronounced increases in DUhTP mice. For example, the platelet-activating factor was significantly increased in trained DUhTP compared with sedentary mice and, in contrast, not changed in control mice (Appendix A). Other phospholipids with signaling functions, such as lysophosphatidylcholines, were increased during physical activity in both mouse groups.

Altogether, there were more lipids in the circulation of trained marathon mice and more glucose and lactate in that of trained control mice, suggesting oxidative lipid metabolism in the muscles of DUhTP and carbohydrate-based metabolism in the muscle of control mice.

### 3.4. Treadmill Running Results in Higher Glycogen Levels and Reduced Lactate Dehydrogenase Activity in Muscle from DUhTP Mice

Higher glycogen content was found in the Mrf of trained DUhTP mice than in the Mrf of sedentary littermates or trained DUC mice (*p* < 0.05; Figure 4a). We further demonstrated that DUhTP mice showed a reduction in lactate dehydrogenase (LDH) activity in response to repeated physical activity (*p* < 0.0001; Figure 4b), whereas treadmill running did not induce any changes in controls. However, the activity of muscle LDH differed by 20% (*p* < 0.0001) between trained animals of both lines. Pyruvate dehydrogenase (PDH) activity, instead, was significantly increased in sedentary DUhTP mice compared with controls (*p* < 0.005; Figure 4c). After treadmill training, a 70% higher PDH activity was observed in DUhTP mice (*p* < 0.05), while DUC mice only showed a 44% increase (not significant). Therefore, PDH enzyme activity differed by 117% between both trained lines (*p* = 0.0001). The enzyme activity of IDH was increased in sedentary and trained DUhTP compared with the groups of controls (Appendix A, *p* < 0.01). CK activity was decreased in trained DUhTP muscles compared with those of trained DUC (*p* < 0.05; Appendix A) and sedentary DUhTP mice (*p* < 0.01).

Thus, the results clearly argue against anaerobic glycolysis (increased glycogen stores as well as reduced LDH activity) and energy supply by CK in the muscle of trained DUhTP mice compared with trained controls; instead, the results suggest increased substrate oxidation shown by increased PDH and IDH activities.

## 4. Discussion

The primary objective of the present study was to identify control mechanisms of lipid-based energy metabolism in the muscle of the non-inbred marathon mouse model (DUhTP) related to their superior running performance in contrast to an unselected control line (DUC), originally derived from the same genetic background. To this purpose, we investigated the interactions between long-term phenotype selection for high treadmill performance (indirect effects) and repeated high-speed treadmill training (direct effects) on muscle metabolism and asked which metabolic mechanisms and adaptations manifested in the muscle through phenotype selection to ensure the exceptional endurance capabilities of DUhTP mice after 140 generations of selection. Data from the current study indicate (i) that long-term selection resulted in metabolic adaptations in the DUhTP mouse line and (ii) that three weeks of treadmill training at intensive speed directly favors energy expenditure from glycolytic utilization in DUC but appears more to rely on lipid-based metabolism in DUhTP mice, with substrate flux controlled by specific enzyme activities.

Phenotype selection resulted in reduced body mass and muscle weight of DUhTP compared with control mice but no significant structural changes in Mrf. Since both muscle mass and body weight were reduced during selection, the ratio of muscle to body mass was the same as in the unselected controls, indicating a general reduction in the size of the DUhTP animals. Based on the significantly increased gene expression of metabolic enzymes involved in glycolysis, the TCA cycle, oxidative phosphorylation, and FAO in the Mrf of marathon mice compared with controls, metabolic adaptations can be assumed as a consequence of phenotypic selection, since these animals have never trained. In particular, the transcript abundance of enzymes related to beta-oxidation and the TCA cycle was, on average, 50% higher (logFC ≈ 0.5) in DUhTP mice than in controls. Consistent with the increased transcript levels of Pdhb, Idh2, and Idh3, both PDH and IDH enzyme activities are increased in the DUhTP mice, suggesting a stronger oxidative metabolic potential of the muscles of the DUhTP mice.

In response to physical activity, significant differences were found between the muscles of both lines and partially between trained and sedentary DUhTP mice. The body and muscle weights of trained DUhTP mice were even lower than those of sedentary littermates. Concomitantly, the CSA of the muscles from trained DUhTP mice was also reduced. Because the Mrfs of the trained DUhTP mice simultaneously had a significantly higher number of muscle fibers per square millimeter of CSA than the Mrfs of trained DUC mice, it is reasonable to speculate that the higher number per square millimeter of CSA is related to the decrease in the cross-sectional area of muscle fibers, particularly the decrease in the area of white muscle fiber area. Thus, in the counted areas in the trained DUhTP mice, and only in them, the number of white fibers correlated negatively with their fiber area, whereas the number of intermediate fibers correlated positively with their area. Overall, more mitochondria-rich fibers, i.e., intermediate and red fibers, were detected per square millimeter of counted area in trained marathon mice, which are generally characterized by smaller diameters than fast glycolytic fibers [38]. Similar observations have been made in mice after six weeks of voluntary running wheel use [39] or in guinea pigs after eight weeks of treadmill training [40]. Thus directly after three weeks of exercise, treadmill running induced a greater shift toward the oxidative fiber type in DUhTP versus DUC mice, which may be associated with a stronger preference toward fat-dependent metabolism and may indicate improved metabolic flexibility [22,41,42] in marathon mice than in unselected control mice in response to running training.

Although metabolic flexibility was not explicitly examined in this study, RNA transcript levels suggest increased oxidation of glucose and lipids through increased glycolysis, beta-oxidation, TCA cycle, and respiratory chain transcripts in the Mrfs of trained DUhTP mice. In particular, transcript levels of hexokinase 2, the initial enzyme of glycolysis and, therefore, an essential player in metabolic flexibility, were induced in both lines by exercise. Notably, increased hexokinase-2 transcript levels in trained mouse skeletal muscle have also been described in a recent review of endurance-associated gain-of-function and loss-of-function genes [43]. In C57BL6 mice running to exhaustion, threefold hexokinase 2 overexpression results in improved endurance performance, and 50% knockout results in decreased endurance performance compared with wild-type mice [44]. Muscle glycogen is not spared in partial knockout animals. During high-intensity exercise, muscle glycogen generally becomes more critical [45], although glycogen-sparing animals have also been reported to have high levels of free fatty acids circulating in their plasma [46]. In particular, with increasing training intensity, glycogen stores are depleted [17] as in Wistar rats by up to 87% in the red vastus lateralis muscle after thirty minutes of treadmill training at 25 m/min [47] or in BALB/c mice by 33% in gastrocnemius muscle after five days of thirty minutes of treadmill training at 25 m/min [48]. Although the treadmill speed in the last week of training was 30 m/min in DUhTP mice, there was no depletion of glycogen stores. This may be due to decreased expression of glycogen-degrading enzymes (Phkb, Agl) and increased expression of glycogen-synthesizing enzymes (Gyg, Gys1). On the other hand, it can be assumed that energy production from plasma glucose occurs without glycogen utilization in contrast to control mice. In trained control mice, blood glucose levels were elevated compared with trained DUhTP mice, the same as blood lactate levels.

Blood lactate levels, as a product and substrate of LDH, were not affected by treadmill running in DUhTP mice and were similar to those of sedentary littermates. Lactate is produced primarily by glycolytic fibers [4,7], although oxidative fibers also produce lactate [49]. It can be assessed as a measure of the anaerobic glycolytic contribution to exercise as a function of exercise intensity, duration, and type of physical activity. For example, peak blood lactate concentrations were observed after a supramaximal 400 m sprint and were thus higher than in long-distance runners or sedentary subjects [50]. Blood lactate concentrations of approximately 6–8 mmol/L, as measured in DUC mice, correspond to moderately elevated lactate levels during exercise and indicate that the DUC animals are not overloaded with the selected training volume in this study. High exercise intensity with predominant anaerobic glycolysis results in levels of 10–15 mmol/L and higher [51]. The lack of increased lactate production in the muscles of trained DUhTP mice may be related to the observed decreased LDH enzyme activity, which we confirmed by lower lactate levels three days after three weeks of treadmill training in DUhTP, but not in DUC mice. Consequently, we assumed that reduced LDH activity abolishes anaerobic glycolysis with concomitant lactate formation and instead drives glucose oxidation in the muscle of trained DUhTP mice. Evidence for this is provided by decreased transcript levels of the mitochondrial lactate transporter Slc16a1 [14], increased PDH enzyme activity, increased transcript levels encoding the mitochondrial pyruvate transporters Mpc1 and 2 [13], enzymes of the TCA cycle and oxidative phosphorylation, and the increased activity of IDH, responsible for the conversion of isocitrate to α-ketoglutarate in the TCA cycle, in DUhTP compared with DUC muscles (Figure 5). Increased PDH and IDH activities in contrast to decreased LDH and CK activities in active DUhTP mice compared with sedentary littermates or physically active DUC mice support this hypothesis of increased oxidative metabolism in physically active DUhTP mice. Since PDH controls the influx of pyruvate derived from carbohydrates into mitochondria, it is thought to play a critical role in metabolic flexibility in muscle [52].

The shift from the glycolytic to the oxidative muscle fiber type not only increases the ability to utilize glucose oxidatively but also increases the ability to burn fat as fuel [41]. During fat oxidation, white adipose tissue, particularly subcutaneous abdominal fat [53], plays a key role in the flexibility of lipid supply and serves as a buffer for lipid flux [54]. Fatty acids, stored as triglycerides in adipose tissue, are transported via the blood circulation to cells with increased energy demand and are taken up by diffusion, fatty acid transport protein (FATP1, Slc27a1), or CD36, which requires the induction of their translocation to the cell membrane [16]. Increased blood concentrations of specific circulating triglycerides (especially TAG 62:16), acylcarnitines, phospholipids, and fatty acid derivatives were found in trained DUhTP mice compared with sedentary littermates as well as trained DUC mice. Circulating fatty acid levels increase during endurance exercise, promoting lipid uptake into muscle and muscle oxidation [45,55]. Therefore, Fernández-Verdejo et al. suggested that elevated lipid availability and increased lipid oxidation during exercise are true measures of metabolic flexibility toward lipids [55]. Consistent with increased fat mobilization, trained DUhTP animals have lower epididymal and posterior subcutaneous fat depots than sedentary ones. While muscular Cd36 transcripts did not differ between trained strains, a higher expression of Slc27a1 was observed in trained DUhTP mice.

It has been shown that cellular uptake of long-chain fatty acids from blood across the endothelial-cell layer via the fatty acid transport proteins FATP3 and FATP4 is induced by VEGF-B and coregulated with mitochondrial genes [56]. Muoio postulated that by high VEGF-B production, fatty-acid-utilizing muscle cells communicate their substrate preference to local vessels to be supplied with these nutrients [57]. This facilitates a shift in energy metabolism towards a reliance on lipids. For example, repeated physical activity activates PGC1α and mitochondrial biogenesis and increases oxidative fat catabolism in muscle. Accordingly, we found significantly increased Slc27a4 (FATP4) and Vegfb expression in DUhTP compared with DUC mice, with higher expression after training. Ppargc1a mRNA expression was also increased in trained DUhTP mice compared with sedentary littermates. Interestingly, Storlien et al. hypothesized that induction of PGC1α (due to exercise) and associated increased mitochondrial biogenesis and insulin sensitivity might enhance the metabolic flexibility of muscle cells to effectively switch between carbohydrate and lipid fuel sources [41].

Mitochondrial uptake of activated fatty acids is mediated by carnitine palmitoyltransferase (Cpt) 1 and 2, and muscular CPTI overexpression results in increased FAO and decreased intramuscular fatty acid esterification [58]. The increased Cpt2 transcripts and increased gene expression of FAO enzymes detected in DUhTP mice indicate increased uptake of fatty acids into mitochondria for β-oxidation and explain the decreased intramuscular triacylglycerol storage in trained animals of both lines. The decreased intramuscular triacylglycerol storage may also be related to reduced plasma insulin levels caused by physical activity in both lines, as insulin promotes muscle lipid uptake and intramuscular fat accumulation. However, the inconspicuous intramuscular lipid storage, combined with the increased gene expression of FAO, TCA cycle, and oxidative phosphorylation enzymes observed here, and the increased activity of IDH, a TCA cycle enzyme, in the trained DUhTP animals, suggests increased mitochondrial fatty acid uptake and subsequent oxidative energy production in these mice (Figure 5).

Despite the adaptation of running time and speed in the training design of this study to the capabilities of the mice, anaerobic glucose utilization plays a more important role in energy supply in the control line than in our marathon mouse model in response to physical activity, as evidenced by higher muscular CK activity, higher LDH activity and elevated plasma lactate levels in DUC than DUhTP mice. High circulating plasma lactate levels suppress the need for lipolysis because glucose and glycogen are abundant [8]. In this context, plasma lactate acts directly as a signal to adipose tissue by binding to the highly expressed receptor GPR81 in adipose tissue and leading to a dose-dependent inhibition of glycerol and FFA release via inhibition of adenylyl cyclase [59,60]. In contrast, GPR81-deficient mice do not show an antilipolytic response to lactate [60]. In marathon mice, the anti-lipolytic signal is absent during exercise due to low plasma lactate levels resulting from reduced LDH activity (Figure 5). Instead, increased fat mobilization from peripheral adipose tissues occurs alongside glucose oxidation without glycogen breakdown contribution, resulting in decreased fat depots (epididymal and posterior subcutaneous).

The present study has several limitations. First, the study was evaluated only for genotype × exercise interactions. To test the concept of enhanced energy metabolism developed by long-term selection for high treadmill running performance in DUhTP mice in the future, we need to consider additional interactions, such as age, sex, or nutrition. Furthermore, additional exercise programs, preferably with defined maximal oxygen consumption, could be used to test the state when the exclusively aerobic glucose metabolism is abandoned. Finally, in this study, we exclusively tested effects directly after a three-week training program. However, examining the animals at different time points during the experiment and at later time points after the last training session may be interesting to study intermediate or chronic training effects.

## 5. Conclusions

Compared with the control model, the results of this manuscript impressively demonstrate that three weeks of treadmill training in long-term-selected marathon mice result in hardly any histological (muscle fiber types) but much more metabolic changes. Paternal selection for high running performance has conferred greater metabolic flexibility to the muscle, allowing simultaneous oxidative utilization of glucose and fatty acids during high-speed exercise. We propose that LDH plays a critical role in this process, as its downregulated activity prevents increased lactate production during endurance exercise and thus selectively provides peripheral energy from fat depots. Acute downregulation of LDH in muscle may play a specific role in improving metabolic flexibility in the muscle of marathon mice or may be required to meet the high energy demands during endurance exercise.

## Figures and Tables

**Figure 1 cells-12-01925-f001:**
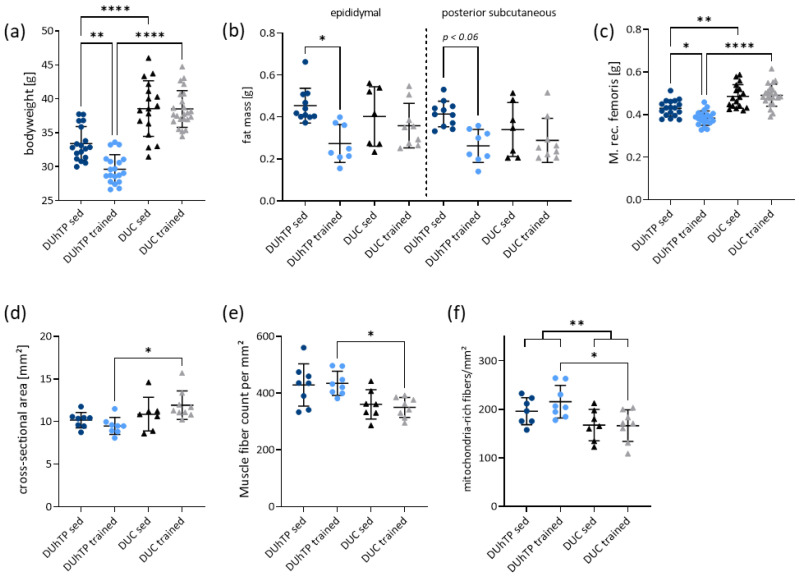
(**a**) Body weight (n ≥ 17), (**b**) epididymal (left panel, n ≥ 7) and posterior subcutaneous fat (right panel, n ≥ 7), and (**c**–**f**) detailed analysis of the muscle in trained and sedentary DUhTP and DUC mice at 70 days of age. (**c**) The dissected *Musculus rectus femoris* (Mrf) were weighted (n ≥ 17), and the (**d**) cross-sectional area (CSA, n ≥ 7), as well as (**e**) muscle fiber count and (**f**) mitochondria-rich fibers per mm^2^ analyzed muscle region (see Method part) were determined. The color used corresponds to the mouse groups (dark blue: DUhTP sed, light blue: DUhTP trained, black: DUC sed, gray: DUC trained). Data are presented as scatter plots with means and standard deviations. Statistical analysis was performed using two-way ANOVA. Significant differences are indicated: * *p* < 0.05, ** *p* < 0.01, **** *p* < 0.0001.

**Figure 2 cells-12-01925-f002:**
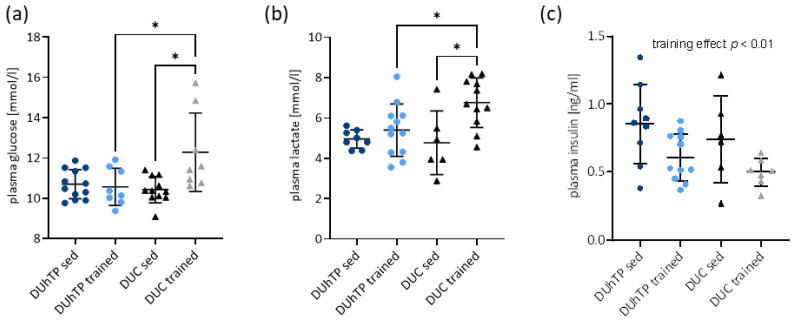
Analysis of metabolites in the circulation. (**a**) Blood glucose (n ≥ 8), (**b**) plasma lactate (n ≥ 6), and (**c**) plasma insulin (n ≥ 6) in trained and sedentary DUhTP (light/dark blue) and DUC mice (gray/black). All results are presented as scatter plots with mean and standard deviations. Statistical analysis was performed using two-way ANOVA. The training-mediated effect on plasma insulin level was statistically evaluated by using an unpaired *t*-test with Welch correction. Significant differences as indicated: * *p* < 0.05.

**Figure 3 cells-12-01925-f003:**
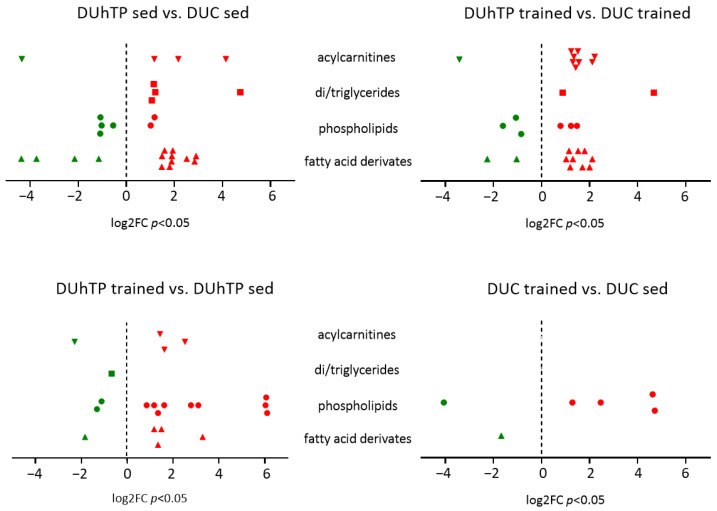
Significantly upregulated (red-colored) and downregulated (green-colored) levels of various acylcarnitines (triangles, tip down), di-/triglycerides (squares), phospholipids (circles), and fatty acid derivates (triangles, tip up) in the serum samples of sedentary or trained DUhTP mice (n = 10) and controls (n = 10) analyzed by untargeted metabolomics using high-resolution LC-MS/MS. Significant (*p* < 0.05) increases or decreases are labeled in red or green. Differences are shown as log2FC.

**Figure 4 cells-12-01925-f004:**
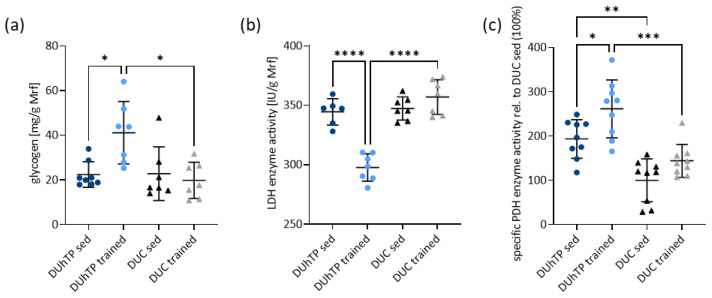
Analysis of metabolites and metabolic enzymes in muscle tissue of trained and sedentary DUhTP (light/dark blue) and DUC mice (gray/black). (**a**) Muscle glycogen content and (**b**) the lactate dehydrogenase (LDH) enzyme activity were determined per gram of *Musculus rectus femoris* (Mrf, n ≥ 6 per group). (**c**) Pyruvate dehydrogenase activity was measured over a duration of 30 min (Mrf, n = 9) and displayed as changes relative to DUC sed. All results are presented as scatter plots with mean and standard deviations. Statistical analysis was performed using two-way ANOVA. Significant differences as indicated: * *p* < 0.05, ** *p* < 0.01, *** *p* < 0.001, **** *p* < 0.0001.

**Figure 5 cells-12-01925-f005:**
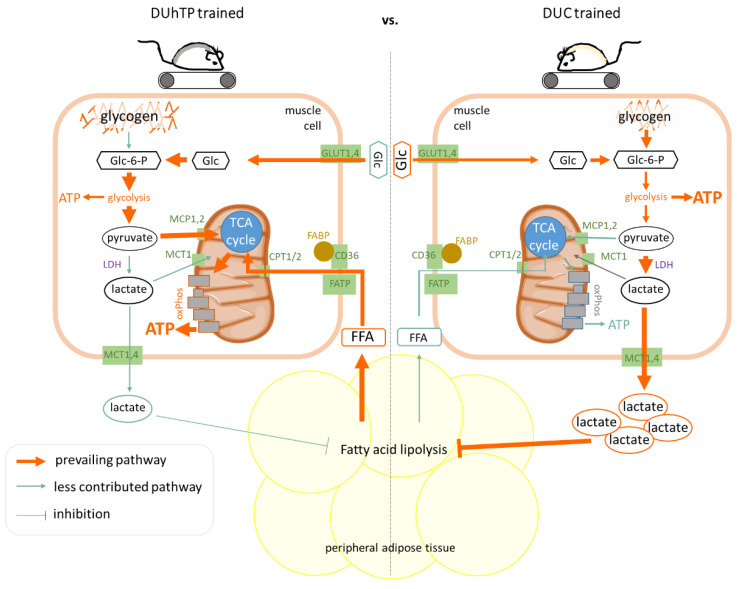
Pathway modeling in *Musculus rectus femoris* (Mrf) of trained DUhTP mice (left) vs. trained DUC mice (right). The underlying data were obtained by NGS, enzyme activity assays, and muscle and plasma levels determination. Dominant pathways are shown in orange, and less-dominant ones are shown in blue-green. Thicker arrows correspond to a higher prevalence. The specific relationships are summarized in the discussion.

**Table 1 cells-12-01925-t001:** List of upregulated (red) and downregulated (green) DEGs involved in metabolic pathways in the four comparison groups extracted from the list of all DEGs (Appendix A). Significant differences as indicated: * FDR < 0.05, ** FDR < 0.01, *** FDR < 0.001, **** FDR < 0.0001, gray cells: comparable gene expression, n = 7–8 animals per group.

Pathway	Involved Genes	Gene Name	DUhTP Sed vs. DUC Sed		DUhTP Trained vs. DUC Trained		DUhTP Trained vs. DUhTP Sed		DUC Trained vs. DUC Sed	
			logFC	FDR	logFC	FDR	logFC	FDR	logFC	FDR
Glycogen metabolism	Phosphorylase b kinase	Phkb	−0.41	**	−0.38	*				
	Glycogen phosphorylase	Pygm			0.44	**				
	Glycogen debranching enzyme	Agl	−0.48	**	−0.35	*				
	Glycogenin	Gyg			0.28	**				
	UTP-glucose-1-phosphate uridylyltransferase	Ugp2			−0.28	*				
	Glycogen synthase	Gys1			0.23	*				
Glycolysis	Glucose transporter Glut1	Slc2a1								
	Glucose transporter Glut4	Slc2a4	0.23	*	0.36	****	0.27	*		
	Hexokinase	Hk2	−0.50	***			0.75	***	0.41	*
	Phosphoglucoisomerase	Gpi1	0.38	***	0.58	****	0.26	0.057		
	Phosphofructokinase	Pfkm			0.30	*				
	Aldolase	Aldoa	0.36	**	0.51	****				
	Phosphotriose isomerase	Tpi1	0.45	****	0.53	****				
	Glyceraldehyde 3-P dehydrogenase	Gapdh	0.66	****	0.84	****	0.39	*		
	Phosphoglycerate kinase	Pgk1	0.29	*	0.27	*				
	Phosphoglycerate mutase	Pgam1			0.73	*				
		Pgam2	0.52	****	0.61	****				
	Enolase	Eno3	0.62	****	0.55	****				
	Pyruvate kinase	Pkm	0.30	*	0.39	***				
	Lactate dehydrogenase	Ldha	0.41	***	0.40	***				
		Ldhd	0.26	*	0.25	*				
	Pyruvate Transporter	Mpc1	0.43	****	0.59	****				
		Mpc2	0.47	****	0.29	**				
	Lactate transporter MCT1	Slc16a1			−0.43	*				
	Lactate transporter MCT2	Slc16a3	0.40	*	0.40	*				
Fatty acid (FA) oxidation	FA transporter CD36	Cd36								
	FA transportprotein 1	Slc27a1	0.58	**	1.03	****				
	FA transportprotein 4	Slc27a4			0.39	***	0.32	*		
	FA binding protein 3	Fabp3	0.54	*	1.14	****				
	Carnitine-O-palmitoyltransferase 2	Cpt2	0.45	**	0.63	****				
	Acyl-CoA dehydrogenase	Acads	0.62	****	0.77	****				
		Acadm	0.52	**	0.53	**				
		Acadl			0.83	****				
		Acadvl	0.57	***	0.84	****				
	Enoyl-CoA hydratase	Hadha	0.32	*	0.68	****				
	Hydroxy acyl-CoA dehydrogenase	Hsd17b10	0.58	****	0.75	****				
	Ketoacyl-CoA thiolase	Acaa2	0.77	****	1.05	****				
TCA cycle	Citrate synthase	Cs	0.43	***	0.50	****				
	Aconitate hydratase	Aco2	0.38	**	0.64	****				
	Isocitrate dehydrogenase	Idh2	0.86	**	1.35	****				
	Isocitrate dehydrogenase	Idh3b	0.42	****	0.55	****				
	Isocitrate dehydrogenase	Idh3g	0.45	****	0.59	****				
	a-ketoglutarate dehydrogenase	Ogdh			0.43	****	0.24	0.062		
	Succinyl-CoA synthetase	Suclg1	0.45	****	0.50	****				
		Suclg2	0.31	**	0.28	**				
		Sucla2	0.27	**						
	Succinate dehydrogenase	Sdhc	0.63	****	0.53	****				
		Sdhd	0.45	****	0.46	****				
		Sdhb	0.47	**	0.72	****				
		Sdha			0.38	***				
	Fumarase	Fh1	0.45	****	0.52	****				
	Malate dehydrogenase	Mdh2	0.53	****	0.83	****				

## Data Availability

All raw data were generated at the FBN Dummerstorf. Derived data supporting the findings of this study are available on request from the corresponding author.

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
