# Peer review of "Metabolic Pathway Modeling in Muscle of Male Marathon Mice (DUhTP) and Controls (DUC)—A Possible Role of Lactate Dehydrogenase in Metabolic Flexibility"

_cells, 2023, doi:10.3390/cells12151925_

Round 1
Reviewer 1 Report (New Reviewer)
In the principal interesting present manuscript it is investigated the contribution of non-mitochondrial and mitochondrial metabolic pathways in response to repeated training in a polygenic, paternally selected marathon mouse model (DUhTP), characterized by exceptional running perfor mance and an unselected control (DUC), with both lines descended from the same genetic background. The study seem well done and the complex results are well presented. It is concluded from the present results that three weeks of treadmill training in long-term-selected marathon mice result in hardly any histological (muscle fiber types) but much more metabolic changes. Moreover that LDH could play a critical role in this process, as its down-regulated activity prevents increased lactate production during endurance exercise and thus selectively provides peripheral energy from fat depots. Some small limitations are given in the specific comments.
Specific comments:
1. It must be well explained on which basis the experimental protocol is developed. Especially the background behind the selection process of the used specific high treadmill performance must be better explained.
2. Is the capilarization in the DUhTP mice also increased as a mechanism for better gas exchange supply which could be from important in case of the alteration of muscle metabolism?
3. I different adaptation of muscle mass in DUhTP mice should be discussed.
Author Response
Reviewer 1:
In the principal interesting present manuscript it is investigated the contribution of non-mitochondrial and mitochondrial metabolic pathways in response to repeated training in a polygenic, paternally selected marathon mouse model (DUhTP), characterized by exceptional running perfor mance and an unselected control (DUC), with both lines descended from the same genetic background. The study seem well done and the complex results are well presented. It is concluded from the present results that three weeks of treadmill training in long-term-selected marathon mice result in hardly any histological (muscle fiber types) but much more metabolic changes. Moreover that LDH could play a critical role in this process, as its down-regulated activity prevents increased lactate production during endurance exercise and thus selectively provides peripheral energy from fat depots. Some small limitations are given in the specific comments.
We are pleased about the positive comments of the Reviewer and thank the Reviewer a lot for the helpful hints and suggestions that have been provided.
Specific comments:
- It must be well explained on which basis the experimental protocol is developed. Especially the background behind the selection process of the used specific high treadmill performance must be better explained.
We thank the Reviewer for pointing out the weakness in our formulations. We have followed the Reviewer's recommendations and briefly summarized the selection process (lines 136-141). We have also rewritten the background of the development of the experimental protocol (lines 158-165).
- Is the capilarization in the DUhTP mice also increased as a mechanism for better gas exchange supply which could be from important in case of the alteration of muscle metabolism?
We have data on capillarization in our mice generated by staining with alkaline phosphatase, according to the method of Spannhof, 1967 (https://books.google.de/books?id=go8PAQAAMAAJ). However, we did not include the results in the manuscript because we found no differences between both lines or in response to physical activity.
To be more specific, we found a similar number of capillaries per fiber in both lines for red, intermediate, and white fibers, with red fibers having more capillaries than intermediate fibers, which in turn had more capillaries than white fibers. Moreover, there were no significant line-specific or training-related differences in the muscle fiber areas supplied by the capillaries. We only found that mitochondria-rich fiber areas (red and intermediate) per capillary were significantly smaller than white fiber areas per capillary.
- The different adaptation of muscle mass in DUhTP mice should be discussed.
Thanks for pointing out that this part is missing from the discussion. We included a discussion on lower muscle mass in DUhTP mice (lines 564-567 and lines 579-590).
Reviewer 2 Report (New Reviewer)
In this manuscript, the authors investigated the metabolic changes in response to physical activities in marathon mice and unselected control. Overall, it's an interesting and well-presented work. However, the following issues need to be addressed before publication.
Major:
1. In the result sections, the authors only listed the findings. There's a lack of a clear and concise summary of each experiment, how it's related to their hypothesis, and how the findings pave the way to the following hypothesis.
Minor:
1. In Figure 1b, why does DUC have a lower fat mass compared with DuhTP?
2. In Figure 1e, the number of myofibers per area was counted. This parameter can't demonstrate the total number of myofibers. Instead, it indicates the size of the myofiber (i.e. CSA of the myofiber) is smaller since more fibers were found in the same area.
Again, the authors didn't mention how to interpret or what to conclude about their findings. It's hard to evaluate if the design of this experiment is appropriate.
3. In Figure 1b, d-f, the significant difference between DuhTP sed vs DUC sed was not labeled or described.
4. The figures should be demonstrated in order. In section 3.3, figure 3 has been discussed. But section 3.4 discussed Figure. 2
Overall, the language is clear and easy to read. There're minor issues that need to be checked carefully and revised. For example, in line 326, it should be "fat" mass instead of "fast" mass.
Author Response
Reviewer 2:
In this manuscript, the authors investigated the metabolic changes in response to physical activities in marathon mice and unselected control. Overall, it's an interesting and well-presented work. However, the following issues need to be addressed before publication.
We are delighted that the Reviewer appreciates our work and thank the Reviewer for the valuable input, which was instrumental in improving the manuscript.
Major:
- In the result sections, the authors only listed the findings. There's a lack of a clear and concise summary of each experiment, how it's related to their hypothesis, and how the findings pave the way to the following hypothesis.
We thank the Reviewer for this suggestion, which we have gratefully implemented. Now, there is a short and clear summary at the end of each result section related to the hypothesis, trying not to discuss the results. The additional sentences are highlighted.
Minor:
- In Figure 1b, why does DUC have a lower fat mass compared with DuhTP?
DUhTP mice generally have higher fat mass than DUC mice, but these differences become significant when fat mass weight is related to body weight (relative fat mass), as published in 2015 (DOI: 10.1159/000442399). In Fig. 1b, we plotted absolute fat mass to demonstrate the fat depot reduction due to exercise, which was exclusively observed in DUhTP mice. The absolute fat masses were not statistically significant between the lines.
The 2015 study addressed this question: "Why do DUhTP mice have higher relative fat mass than DUC mice under sedentary conditions, despite long-term selection for high running performance?" Based on the results of that study, we hypothesized that DUhTP mice acquired a set of energetic-metabolic adaptations in subcutaneous fat during long-term selection to fuel high energy demands during physical activity. This includes the ability to break down accumulated fat mass even during moderate physical activity. With the results of this study presented here, we have been able to extend the hypothesis further. Here, we now show that during physical activity, lactate production in the DUhTP mice is kept low, which in turn drives lipolysis in adipose tissue.
In Figure 1e, the number of myofibers per area was counted. This parameter can't demonstrate the total number of myofibers. Instead, it indicates the size of the myofiber (i.e. CSA of the myofiber) is smaller since more fibers were found in the same area.
Again, the authors didn't mention how to interpret or what to conclude about their findings. It's hard to evaluate if the design of this experiment is appropriate.
We agree with the Reviewer that it is not possible to reflect the total number of all myotubes. This is not possible at all. Instead, we used two visual fields, each in the superficial and in the profound regions, for counting. These counted fibers were then related to the counted area. We also agree with the Reviewer that the higher number of fibers per area in the muscles of trained DUhTP animals is related to the reduced fiber cross sections.
We rechecked our data and the text and revised the method description, result part, and discussion. We removed the wrong phrase, 'the total number of muscle fibers per cross-section,' and rewrote the passage (lines 182-194), which is now reading: 'Immunohistochemistry was performed on the Musculus rectus femoris (Mrf) of seven (DUC sed) to eight animals per group (DUhTP sed, DUhTP trained, DUC trained), as described by Rehfeldt et al. [32]. Briefly, 10 µm cross-sections were cut at the end of the proximal origin tendon, and the metabolic fiber types (red, intermediate, white) were stained using NADH-tetrazolium reductase. The muscular cross-sectional area (CSA) and the fibers over an area of 3.316 mm² (approximately 32 % of the total CSA) were determined separately by image analysis (TEMA v1.00, Scan Beam APS, Hadsund, Denmark). In these analyzed areas, a total of four fields of view (representative images for one animal per group are shown in Fig. S2c), 100-150 muscle fibers each in the light (superficial) and dark (profound) areas were evaluated for the distribution and CSA of the red, intermediate, and white fibers.' Furthermore, we corrected the Result part by adding the phrase 'per mm² analyzed muscle region '(see lines 351, 358, 387).
- In Figure 1b, d-f, the significant difference between DuhTP sed vs DUC sed was not labeled or described.
We rechecked the text and our statistics and can confirm that there are no significant differences between sedentary DUhTP and sedentary DUC mice concerning fat mass (fig. 1b), cross-sectional Mrf area (fig. 1d), muscle fiber count per mm² (fig. 1e), and mitochondria-rich fibers/mm² (fig. 1f). Not significant differences were not labeled in the figures. In the text, we stated that there are similar cross-sectional areas (lines 347-348) and no differences in fiber count (351-353).
However, in this context, we found a spelling mistake in Figure 1e that we corrected (fiber count instead of fibre count).
- The figures should be demonstrated in order. In section 3.3, figure 3 has been discussed. But section 3.4 discussed Figure. 2
We agree with the Reviewer and have therefore generated figures 4a-c from figures 2d-f. Consequently, figure 4 became figure 5. The changes in numbering and the legends are highlighted.
This manuscript is a resubmission of an earlier submission. The following is a list of the peer review reports and author responses from that submission.
Round 1
Reviewer 1 Report
The manuscript from Brenmoehl et al brings an expressive amount of data related to mice selected for their ability to perform prolonged aerobic exercise. I have made some comments and questions to the authors hoping to contribute to their careful and interesting work.
This reviewer understood that the main hypothesis of this work was that marathon mice would metabolize lipids more efficiently than the control mice. Objectives were described only in the Discussion section.
The statistical methods used do not seem the most appropriated. Why one-way ANOVA was used for some parameters and two-way ANOVA for other ones? Two-way ANOVA seems to be more adequate. Furthermore, why data is presented as non-parametrical (boxplots) if parametrical tests were used? If data is parametrical, they should be represented by mean and standard error (bar graphics, preferable showing all points, as each analysis had a different number of animals per group). How were data distribution analyzed?
Seems that the next generation sequence (NGS) results were not validated. Could the authors clarify if the metabolites identified were able to validate the NGS results?
Table 1 could be placed as a supplemental material and a heatmap graphic could bring these results to the main text (this is just a suggestion).
Figures – data are presented as non-parametrical, but parametric test were used in the statistical analysis.
LDH activity was lower in the DUhTP trained group, which also presented lower plasma lactate compared to the DUC trained group. These results seem to be caused by the mice selection that favored those with muscle with the highest number of oxidative fibers. Would authors agree with that?
Reviewer 2 Report
Brenmoehl et al. aimed to describe metabolic flexibility in muscles obtained from selected high-capacity runners’ mice. The authors used DUhTP mice, a mouse model of selected high endurance runners, and compared them with non-selected mice in a sedentary state or after a 3 wks period of training. The study failed to address the main question whereas the experimental design and the methodology do not test the hypothesis directly. Also, there is an overall overinterpretation of the results as well as a lack of experimental data to support the claims. Please find my specific comments below which I hope will be useful for the authors.
Majors
1 – The use of one-way ANOVA is not adequate based on the experimental design. The authors should reassess all data using two-way ANOVA comparing genotype vs. training differences. As a result, changes in statistical differences should be followed by a reinterpretation of the data.
2 – The authors used FDR <0.05 in the DEG analysis. However, given the number of variables, this approach leads to an increase in false-positive findings. Therefore, data should be analyzed, and interpreted, using FDR <0.01.
3 – The authors should provide a better characterization of the phenotype, before and after training. Even though this model has been published before, it is important to guarantee internal controls, for instance, maximal speed capacity, and food intake, among others, should be provided.
4 – One major concern is the experimental design of the study. The authors claim in the entire study the effects of training; however, mice were killed after the last bout of exercise, which totally masks the effects of chronic training. Therefore, the whole interpretation of the study as well as the conclusions are, in my opinion, wrong. To name one but not the only one, in line 306 the authors state “total fat content was decreased by training” whereas it is known that fat mobilization/oxidation is very important for exercising muscle, and as I mentioned, this is not an effect of training but rather the last exercise bout.
5 - The authors did not provide any evidence of the classical effects of training (greater mitochondrial content) which makes me wonder whether 3 weeks of 30 min/day is sufficient to elicit chronic training adaptations in these mice, regardless of the genotype. Moreover, the authors did not provide any data showing the effects of the last section of exercise in the studies population (i.e. metabolites ATP, ADP, AMP, PCr, and creatine must be measured as well as phosphorylation levels of AMPK).
6 – The authors mentioned in the title, abstract, and throughout the entire paper the term “metabolic flexibility”. Metabolic flexibility is the ability of the body, mainly skeletal muscle, to change from one substrate to another. There is no single experiment showing any effects of training status or genotype that relates to metabolic flexibility.
7 – Statistical analysis performed in figure S1 is not straightforward to interpret. If the authors wanted to compare the CSA of muscle fibers across groups (genotype/exercise), it is not necessary to compare within the same group across different muscle types. However, if the authors want to keep this approach, an ANCOVA should be used to test such differences.
8 – The whole paper is based on sequencing data of metabolism-related genes whereas no direct evidence of lipid/glucose oxidation is provided. As mentioned earlier, the study suffers from overinterpretation of the data, assuming greater/lower metabolic pathways based on DEG data. Notably, authors calculate LDH/IDH ratio as a surrogate of oxidative capacity, which is physiologically meaningless since the abundance of those enzymes is not comparable as well as they are not key rate-limiting enzymes for fuel utilization in skeletal muscle. The study would benefit from the measurement of PDH activity, ETC complexes activity, and mitochondrial respiration using several substrates.
Minors
9 – Data should be presented with individual observations instead of bars with minimum and maximum.
10 – Representative histological images of skeletal muscles should be provided.
11 – What do the authors mean by mitochondria-positive fibers? Does this mean that there are fibers negative for mitochondria? Also, WB images do not show differences between the groups, so I suggest the authors choose a better representative image. Lastly, the authors state an increase in complex IV, which is almost impossible to see in the WB images, probably due to the boiling of the samples described in the methods section. Boiling samples result in the disappearance of complex IV, as described in the datasheet of the Abcam antibody. Therefore, I suggest excluding the complex IV analysis from the WB since the results are not reliable.
12 - It is not described how intramuscular lipids were measured. Please, add this to the methodology section.
13 - Line 43 – not only intensity and duration but also nutrition, sex, and age affect fuel selection. Please, be specific in the statements.
14 - Line 50 – What do the authors mean by “inducible GLUT4”? Do you mean the translocation of GLUT4 to the plasmatic membrane? Please, be specific when it comes to terminology.
15 - Line 52 – glycogen is also stored in oxidative muscles. Please, change accordingly.
16 – Line 54 – please, a reference is missing with this statement.
17 – Apart from Cori’s cycle, very little evidence exist regarding the direct capacity of mitochondria to utilize lactate whereas experimental evidence, mainly from muscle, does not support this theory (Yoshida et al. 2007). Therefore, I suggest the authors be careful with strong statements when the evidence is not so solid and rewrite the paragraph (line 50) to avoid misleading the readers.
18 – Third and fourth paragraphs in the introduction present excessive redundant and poorly written sentences. I suggest the authors rewrite these paragraphs to be clear about the messages they want to pass along.
19 - How do the authors explain glucose blood levels above 10 mM and lactate levels above 4 mM in a sedentary state?
20 – Lines 490-493 “An exercise-triggered switch from glycolytic to oxidative fiber type, as seen here in trained DUhTP in contrast to trained DUC mice, or vice versa, is accompanied by a transition between carbohydrate- and fat-dependent metabolism and suggests higher metabolic flexibility”. This sentence does not read right and should be rewritten.